# Atopic dermatitis and cognitive dysfunction in middle-aged and older adults: A systematic review and meta-analysis

Qi Zhou[1]*, Dejiang Yang[2], Chongyu Xiong[1], Xinming Li[2]

1 Department of Neurology, The First People's Hospital of Fuzhou, Fuzhou, Jiangxi, China, 2 Department of Neurology, Nanchang First Hospital, Nanchang, Jiangxi, China

* 17807026495@163.com

**Data Availability Statement:** All relevant data are within the paper and its Supporting information files.

**Funding:** The author(s) received no specific funding for this work.

## Abstract

### Background

Atopic dermatitis (AD) is a common chronic inflammatory skin disease that affects adults worldwide. Recent evidence suggests that AD may be associated with cognitive dysfunction, but the results of individual studies have been inconsistent. This systematic review and meta-analysis aimed to evaluate the association between AD and cognitive dysfunction in middle-aged and older adults.

### Methods

To find relevant research, a comprehensive search of electronic databases from the beginning to March 2023 was carried out. Data were taken from studies that were eligible, and a meta-analysis was done to determine the pooled hazard ratio (HR) and 95% confidence interval (CI).

### Results

We searched three databases and found a total of 15 studied arms included in 5 cohort studies with over 8.5 million participants were included in the analysis. The results showed that individuals with AD had a higher risk of developing dementia of all-cause dementia (pooled hazard ratio (HR) = 1.16; 95% CI, 1.10–1.23,P<0.001) and the Alzheimer type (pooled HR = 1.28; 95% CI, 1.01–1.63,P<0.001) but not vascular dementia (pooled HR = 1.42; 95% CI, 0.99–2.04,P<0.001). Subgroup analyses showed that the association between atopic dermatitis and all-cause dementia was significant in Europe (P = 0.004) but not in Asia (P = 0.173) and was significant in prospective cohort studies (P<0.001) but not in non-prospective cohort studies (P = 0.068). Sensitivity analysis and publication bias detection confirmed the reliability of the overall findings.

### Conclusions

In conclusion, this study demonstrated that AD was associated with increased risk of cognitive dysfunction, particularly dementia of the Alzheimer type and all-cause dementia, in

**Competing interests:** The authors have declared that no competing interests exist.

middle-aged and older participants. Further research is needed to understand the mechanisms behind this association and its potential implications for clinical practice.

## Systematic review registration

PROSPERO, identifier (CRD42023411627).

## Introduction

Atopic dermatitis (AD), also known as eczema, is a common chronic inflammatory skin disease that affects a significant proportion of the population worldwide. It is characterized by dry, itchy, and inflamed skin, and can significantly impact patients' quality of life. Recent studies have suggested that there may be a potential link between AD and cognitive dysfunction in the elderly population, including all-cause dementia, dementia, and mild cognitive impairment [1, 2].

Cognitive dysfunction is a decline in cognitive function, including memory, attention, and language abilities. It is a significant public health concern, particularly in the elderly population, as it can lead to impairments in daily functioning, increased dependency, and reduced quality of life [3]. Several studies have reported a higher of cognitive dysfunction among individuals with AD compared to those without the condition [1, 2, 4].

Despite these findings, these findings did not resolve the controversy over the molecular mechanism between AD and cognitive dysfunction. Some studies have suggested that AD may directly contribute to cognitive impairment via neuroinflammation and oxidative stress, which are associated with AD pathophysiology [5]. Others have proposed that the association may be mediated by shared risk factors such as age, genetic vulnerability, and lifestyle factors. For instance, sleep disturbance, depression, and anxiety are common comorbidities of AD, and have been associated with cognitive dysfunction [6].

This systematic review and meta-analysis aim to provide a comprehensive overview of the current evidence on the association between AD and cognitive dysfunction in the in middle-aged and older adults. By synthesizing the results of existing studies, we hope to clarify the nature of the relationship between these two conditions and identify potential avenues for future research. Specifically, we aim to answer the following research questions: (1) What is the strength of the association between AD and cognitive dysfunction in the in middle-aged and older adults? (2) What are the potential mechanisms underlying this association? (3) What are the implications of this association for clinical practice and public health?

## Methods

### Protocol and registration

This systematic review and meta-analysis were conducted following the Preferred Reporting Items for Systematic Reviews and Meta-analyses (PRISMA) statement [7] (S1 Table). This study project has been registered in PROSPERO with an ID number of CRD42023411627.

### Search strategy

A comprehensive search was conducted in the following electronic databases: PubMed, Embase, and Web of Science. The search was conducted from the inception of each database to March 2023. The search strategy was developed using a combination of Medical Subject Headings (Mesh) and free-text terms related to "atopic dermatitis", "dementia", "Alzheimer's

diseases", "vascular dementia", "middle-aged" and "older adults", and "cohort studies" which is shown in S2 Table. The search was limited to human studies. We only included studies that were published in English. We also conducted a manual search of the reference lists of the papers we identified through electronic searching to identify additional studies.

## Inclusion criteria and exclusion criteria

Studies were included in this systematic review and meta-analysis if they met the following criteria: (1) the study design was cohort or case-control; (2) the study investigated the association between atopic dermatitis and cognitive dysfunction in the middle-aged and older adults; (3) the study reported quantitative data on the risk of cognitive dysfunction in the middle-aged and older adults with atopic dermatitis; (4) the study provided a control group of middle-aged and older adults without atopic dermatitis; and(5) the studies provided the effect size hazard ratio [HR], with 95% confidence interval [CI]. Studies were excluded if they: (1) reported outcomes other than the risk of cognitive dysfunction in the middle-aged and older adults; (2) were duplicates, case reports, or reviews; (3) were conducted on animal models or in vitro; (4) did not provide sufficient data for meta-analysis; (5) were conducted on individuals with other dermatological diseases. We defined cognitive dysfunction as a category of cognitive impairment diseases that met the DSM-5 criteria for various subtypes of cognitive disorders, including all-cause dementia, Alzheimer's disease dementia, vascular dementia, mild cognitive impairment [8]. We defined middle-aged adults as those between 45 and 59 years old, while older adults as those aged 60 years and above [9]. Two authors (X.L. and D.Y.) independently screened titles and abstracts initially and then evaluated full-text articles to ensure the included studies met the eligible inclusion criteria. Any disagreement between them was settled by another author (Q.Z.).

## Selection of studies and data extraction

Two reviewers (Q.Z. and C.X.) independently screened the titles and abstracts of all identified studies. Full texts of potentially eligible studies were retrieved and examined. Any discrepancies between the reviewers were resolved by discussion and consensus. Data on the risk of cognitive dysfunction in middle-aged and older adults' individuals with atopic dermatitis and the control group were extracted from the included studies. The other data were extracted from each included study, including study characteristics, participant characteristics, outcome measures, and effect estimates. If data were missing, we contacted the corresponding author for more information.

## Quality assessment of the included studies

The Cochrane Non-randomized Studies Methods Working Group recommended the use of the NOS to assess the quality of observational studies (range: 0–9 stars) [10]. According to the score stars of the NOS, the included studies were deified as low- (1–3 stars), moderate- (4–6 stars), and high quality (7–9 stars). Therefore, if the study obtained ≥4 stars, it was considered to have an above-moderate quality and, thus, was incorporated into our meta-analysis. Data extraction and quality assessment were conducted by two independent investigators (X.L. and D.Y.), and disagreements between them were resolved through negotiation with a third researcher (Q.Z.).

## Data synthesis and analysis

Meta-analysis was performed to estimate the pooled effect size and its 95% CI using the DerSimonian and Laird random-effects model, which assumes that the true effect size varies across

studies due to both within-study and between-study variation. We used the inverse variance method to weight each study according to its precision, which is inversely proportional to the variance of its effect size estimate. The $I^2$ statistic was used to assess the heterogeneity among studies. $I^2$ values of <25%, 25–50%,51–75%, and >75% were considered to denote no, mild, moderate, and large heterogeneity, respectively [11]. Sensitivity analyses were performed to validate the stability of pooled HRs of cohort literature by removing individual study. A p-value <0.05 was significant. The Stata program was used for all analyses (version 17.0; Stata SE Company LP, College Station, TX, USA).

## Results

### Baseline characteristics of the studies

Our initial search identified 1853 articles, and after reviewing the titles and abstracts, 48 articles were selected for full-text screening. Finally, 15 studied arms included in 5 articles met our inclusion criteria and were included in our systematic review and meta-analysis (S3 Table). A total of 8,595,252 participants with atopic dermatitis and corresponding controls were included in the 5 articles. The participants in the 5 articles ranged in mean age from 45 to 74 years. The duration of follow-up ranged from 8.1 to 12 years. The prevalence of atopic dermatitis among the elderly varied widely across studies, ranging from 0.4% to 12%. The search flow chart is shown in Fig 1.

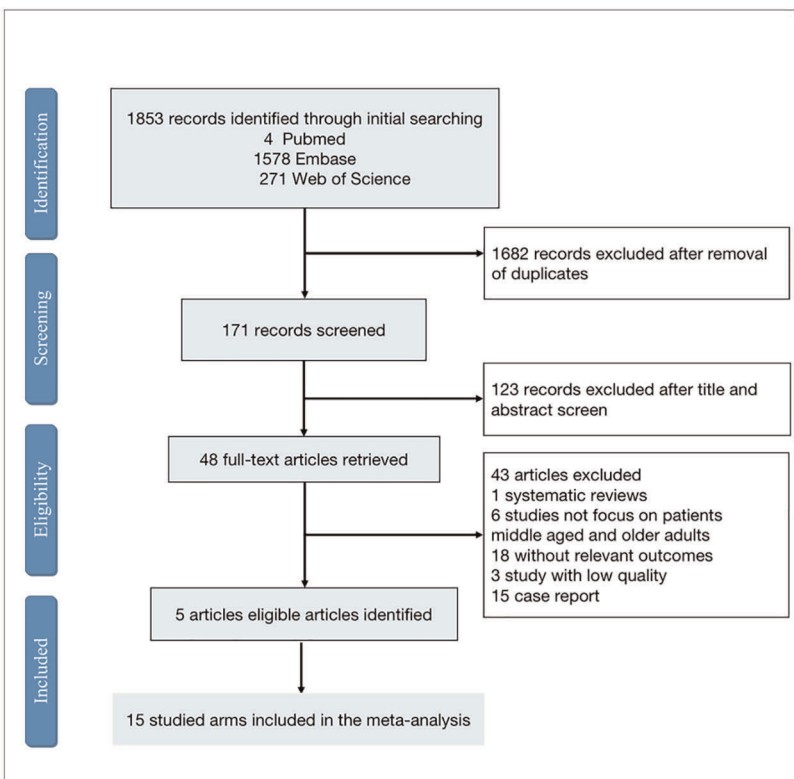

**Fig 1. Flowchart displaying the choice of study.**

### Quality evaluation

To determine the likelihood of bias in the chosen studies, we applied the NOS assessment. As S4 Table demonstrates, the meta-overall analysis's evidence quality was strong.

### Atopic dermatitis and incident of cognitive dysfunction in middle-aged and older participants

Five studied reported effect measures for the association between atopic dermatitis and incident of cognitive dysfunction in middle-aged and older participants were included in quantitative analyses. Six studied arms examined the association between atopic dermatitis and all-cause dementia with a pooled HR of 1.16 (95% CI, 1.10–1.23, p < 0.001; Fig 2) and with substantial heterogeneity across the studied arms ($I^2$ = 54.5%, P = 0.052; Fig 2). Atopic dermatitis was also found to have a significant association with dementia of the Alzheimer type (HR 1.28, 95% CI, 1.01–1.63, p < 0.001; Fig 2). The heterogeneity among the studies was large ($I^2$ = 95.8%, p < 0.001; Fig 2). However, no significant association was found between atopic dermatitis and vascular dementia, with a pooled HR of 1.42 (95% CI, 0.99–2.04, p < 0.001; Fig 2), with significant heterogeneity ($I^2$ = 96.0%, p < 0.001; Fig 2).

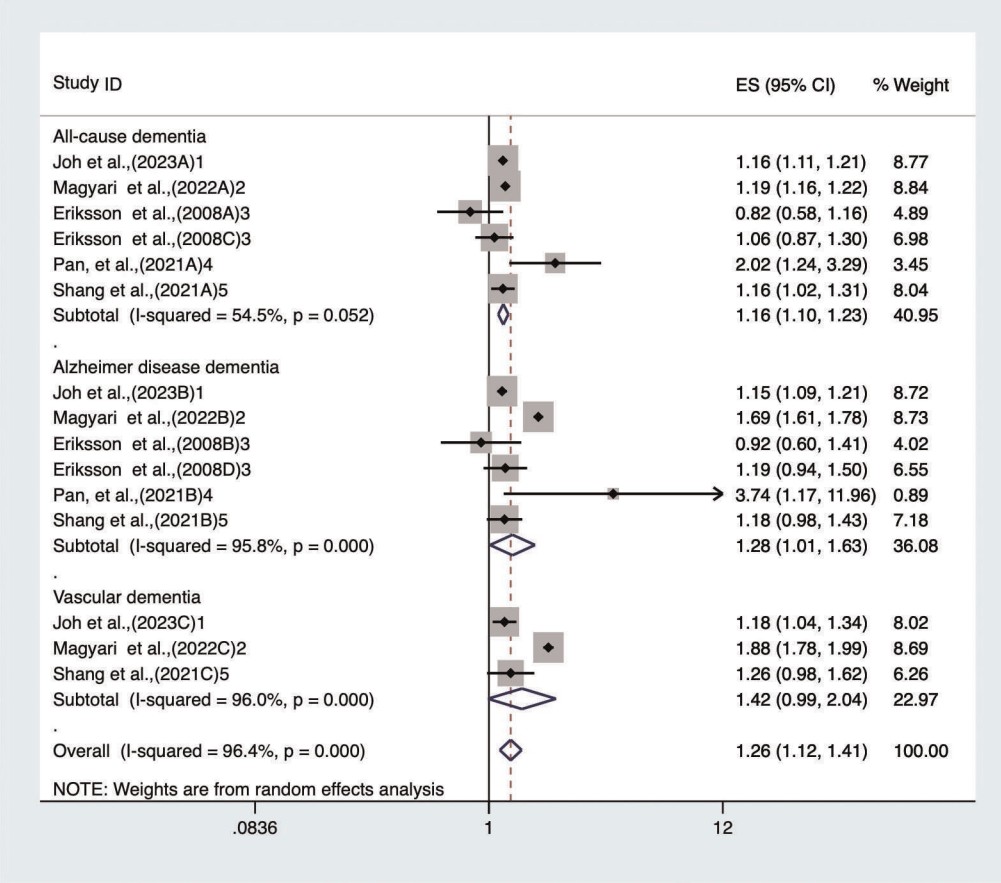

**Fig 2. The forest plot shows the pooled HR for all-cause dementia, Alzheimer's disease dementia and vascular dementia in atopic dermatitis vs non-atopic dermatitis.**

## Sensitivity analysis and publication bias detection

Sensitivity analysis was conducted to evaluate the impact of excluding studies with a high risk of bias on the overall pooled HR for all-cause dementia, Alzheimer's disease-related dementia, and vascular dementia, respectively. We assessed the quality for each study using the Newcastle-Ottawa Quality Assessment Scale. Based on this criterion, we excluded one study of Joh et al. with a score of 6 on the NOS scale from our meta-analysis. The result of the sensitivity analysis showed that the exclusion of this study did not change the direction or significance of the pooled HR for any outcome, seen Fig 3. Due to each group having fewer than 10 studied arms, a funnel plot and Egger's test was not utilized to evaluate a potential publication bias.

## Subgroup analysis

Subgroup analysis was conducted to explore the potential sources of heterogeneity. The subgroup analysis by region showed that the association between atopic dermatitis and all-cause dementia was significant in Europe (HR 1.14, 95% CI, 1.04–1.24, P = 0.004; $I^2$ = 46.5%), but not Asia (HR 1.45, 95% CI, 0.85–2.47, P = 0.173; $I^2$ = 79.7%) (S1 Fig). The subgroup analysis by study design showed that the association between atopic dermatitis and all-cause dementia was significant in prospective cohort study (HR 1.18, 95% CI, 1.12–1.24, p < 0.001; $I^2$ = 55.5%), but not on non-prospective cohort study (HR 1.02, 95% CI, 0.73–1.41, P = 0.068; $I^2$ = 70%) (S2 Fig). The subgroup analysis by region showed that the association between atopic dermatitis and Alzheimer's disease-related dementia was not significant in both Asia (HR 1.79, 95% CI, 0.58–5.47, P = 0.309; $I^2$ = 74.6%) and Europe (HR 1.26, 95% CI, 0.96–1.66, P = 0.101; $I^2$ = 88.9%) (S3 Fig). The subgroup analysis by study design showed that the association between atopic dermatitis and Alzheimer's disease-related dementia was significant in

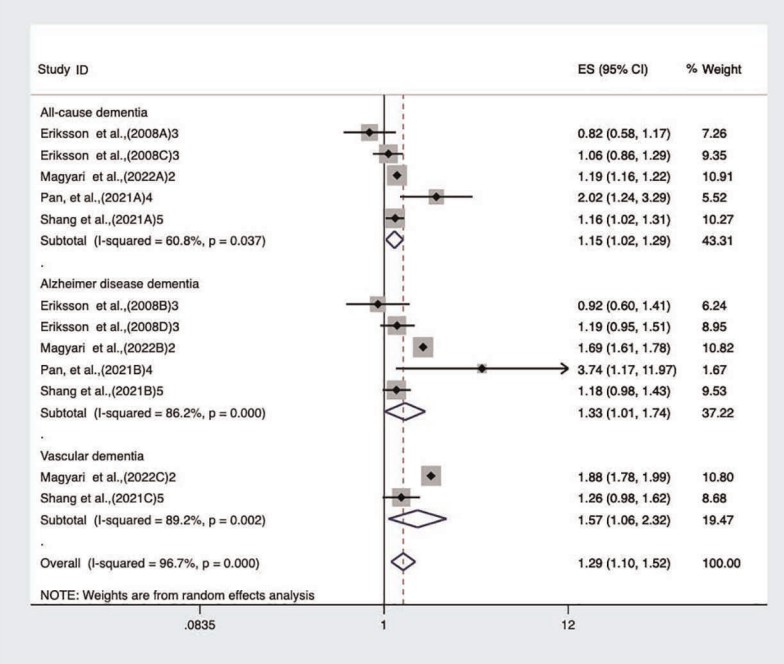

**Fig 3. The sensitivity analyses performed to validate the stability of pooled HRs of cohort literature by removing Joh et al. study for (A) all-cause dementia, (B) Alzheimer's disease dementia and (C) vascular dementia.**

prospective cohort study (HR 1.41, 95% CI, 1.04–1.90, P = 0.025; $I^2$ = 97.3%), but not in non-prospective cohort study (HR 1.13, 95% CI, 0.93–1.36, P = 0.229; $I^2$ = 8.3%) (S4 Fig). The sub-group analysis by region showed that the association between atopic dermatitis and vascular dementia was significant in both Asia (HR 1.18, 95% CI, 1.04–1.34, P = 0.010) and Europe (HR 1.57, 95% CI, 1.06–2.32, P = 0.024; $I^2$ = 89.2%) (S5 Fig). The subgroup analysis by study design showed that the association between atopic dermatitis and vascular dementia was not significant in both prospective cohort study (HR 1.49, 95% CI, 0.95–2.36, P = 0.084; $I^2$ = 97.7%) and non-prospective cohort study (HR 1.26, 95% CI, 0.98–2.04, P = 0.71; $I^2$ =) (S6 Fig).

Because the amount of data included in the study is insufficient, we cannot use regression analysis to analyze the heterogeneous sources of various research results.

## Discussion

The present study found that AD was associated with increased risk of cognitive dysfunction, especially dementia of the Alzheimer type and all-cause dementia, in middle-aged and older participants. This finding is consistent with previous studies that reported cognitive impairment in adults and children with AD [1, 2, 12]. The possible mechanisms underlying this association may include chronic inflammation, oxidative stress, sleep disturbance, psychological distress, and reduced quality of life [13].

The meta-analysis also revealed that the association between AD and all-cause dementia was significant in Europe but not Asia. This regional difference may reflect the variations in genetic susceptibility, environmental factors, diagnostic criteria, and treatment options for AD and dementia across different populations [14]. Further studies are needed to explore the potential moderators and mediators of this association in different regions.

The study design was another factor that influenced the association between AD and cognitive dysfunction. The prospective cohort studies showed a significant association, while the non-prospective cohort studies did not. This may suggest that prospective cohort studies are more reliable and valid to assess the causal relationship between AD and cognitive dysfunction, as they can reduce the recall bias, selection bias, and confounding factors [15]. However, prospective cohort studies are also limited by their long duration, high cost, and loss to follow-up [16]. Therefore, a combination of different study designs may be necessary to provide a comprehensive and accurate picture of the association between AD and cognitive dysfunction.

This study explores the association between atopic dermatitis (AD) and cognitive dysfunction and has important implications for clinical practice and public health. We found that AD may be a potential risk factor for cognitive dysfunction, especially dementia of the Alzheimer type and all-cause dementia, in middle-aged and older adults. Therefore, clinicians should be aware of this association and screen patients with AD for cognitive impairment using validated tools, such as the Mini-Mental State Examination (MMSE) or the Montreal Cognitive Assessment (MoCA). Moreover, we discussed the possible mechanisms underlying the association between AD and cognitive dysfunction, such as chronic inflammation, oxidative stress, sleep disturbance. Therefore, clinicians should adopt a holistic approach to manage patients with AD and cognitive dysfunction, not only treating their skin symptoms but also addressing their systemic and psychological comorbidities. Finally, we emphasized the need for further research on the causal mechanisms, potential biomarkers, and effective interventions for the association between AD and cognitive dysfunction.

Our study has several limitations that should be considered when interpreting the results. First, we included a small number of studies in our meta-analysis, which may limit the generalizability and precision of our findings. Second, these studies were conducted in the United Kingdom, Taiwan, South Korea, and Sweden, which may not reflect the prevalence and impact

of atopic dermatitis and cognitive dysfunction in other regions or countries. Future studies should include more diverse and representative samples from different ethnicities and regions, to improve the generalizability and accuracy of the results across different populations. Third, we could not assess the causal relationship between atopic dermatitis and cognitive impairment, as most of the studies were observational and subject to residual confounding and reverse causation. Fourth, several factors may contribute to the heterogeneity observed in our meta-analysis. For example, the studies varied in their design, such as prospective cohort, retrospective cohort, and cross-sectional studies, which may have different biases and confounding effects. Furthermore, the studies included different populations of middle-aged and older adults, with different geographic locations, ethnicities, and socioeconomic status. In addition, the studies used different definitions of atopic dermatitis and cognitive dysfunction, which may affect the validity and comparability of the results. Therefore, our results should be interpreted with caution and confirmed by future studies with larger sample sizes, more rigorous designs, and more comprehensive adjustments for confounding factors.

## Conclusions

In conclusion, our meta-analysis suggests that atopic dermatitis is associated with an increased risk of all-cause dementia and Alzheimer's disease-related dementia in middle-aged and older adults. However, there is substantial heterogeneity across the studied arms that needs to be further explored. More high-quality studies are needed to confirm our findings and to elucidate the underlying mechanisms linking atopic dermatitis and cognitive impairment.

## Supporting information

**S1 Table. PRISMA checklist.**
(DOCX)

**S2 Table. Search terms included for each library search.**
(DOCX)

**S3 Table. Overview of studies on the associations between atopic dermatitis and cognitive outcomes in included studies.**
(DOCX)

**S4 Table. Assessment of study quality.**
(DOCX)

**S1 Fig. The subgroup analysis by region showed that the association between atopic dermatitis and all-cause dementia.**
(TIF)

**S2 Fig. The subgroup analysis by study design showed that the association between atopic dermatitis and all-cause dementia.**
(TIF)

**S3 Fig. The subgroup analysis by region showed that the association between atopic dermatitis and Alzheimer's disease dementia.**
(TIF)

**S4 Fig. The subgroup analysis by study design showed that the association between atopic dermatitis and Alzheimer's disease dementia.**
(TIF)

**S5 Fig. The subgroup analysis by region showed that the association between atopic dermatitis and vascular dementia.**
(TIF)

**S6 Fig. The subgroup analysis by study design showed that the association between atopic dermatitis and vascular dementia.**
(TIF)

## Acknowledgments

We would like to thank Nanchang University for providing us with access to a series of available online databases.

## Author Contributions

**Conceptualization:** Qi Zhou.

**Data curation:** Qi Zhou.

**Formal analysis:** Qi Zhou, Dejiang Yang, Xinming Li.

**Funding acquisition:** Qi Zhou.

**Investigation:** Qi Zhou, Dejiang Yang, Xinming Li.

**Methodology:** Qi Zhou, Dejiang Yang, Xinming Li.

**Resources:** Qi Zhou, Dejiang Yang, Chongyu Xiong, Xinming Li.

**Supervision:** Qi Zhou.

**Validation:** Qi Zhou, Dejiang Yang, Chongyu Xiong, Xinming Li.

**Visualization:** Qi Zhou, Dejiang Yang, Xinming Li.

**Writing – original draft:** Qi Zhou.

**Writing – review & editing:** Qi Zhou.

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
