## [Decision Letter · Decision Letter 0]

12 Jul 2023

PONE-D-23-12015Atopic dermatitis and cognitive dysfunction in middle-aged and older adults: A systematic review and meta-analysisPLOS ONE

Dear Dr. Zhou,

Thank you for submitting your manuscript to PLOS ONE. After careful consideration, we feel that it has merit but does not fully meet PLOS ONE’s publication criteria as it currently stands. Therefore, we invite you to submit a revised version of the manuscript that addresses the points raised during the review process.

We look forward to receiving your revised manuscript.

Kind regards,

Shuo-Yan Gau

Academic Editor

PLOS ONE

3. Please upload a new copy of Figure 3 as the detail is not clear. Please follow the link for more information: https://blogs.plos.org/plos/2019/06/looking-good-tips-for-creating-your-plos-figures-graphics/" https://blogs.plos.org/plos/2019/06/looking-good-tips-for-creating-your-plos-figures-graphics/

Reviewers' comments:

Reviewer's Responses to Questions

**Comments to the Author**

1. Is the manuscript technically sound, and do the data support the conclusions?

Reviewer #1: Partly

Reviewer #2: Yes

Reviewer #3: Partly

2. Has the statistical analysis been performed appropriately and rigorously? 

Reviewer #1: No

Reviewer #2: Yes

Reviewer #3: Yes

3. Have the authors made all data underlying the findings in their manuscript fully available?

Reviewer #1: Yes

Reviewer #2: Yes

Reviewer #3: Yes

4. Is the manuscript presented in an intelligible fashion and written in standard English?

Reviewer #1: No

Reviewer #2: Yes

Reviewer #3: Yes

5. Review Comments to the Author

Reviewer #1: This study presents novel perspectives regarding the association between atopic dermatitis and cognitive impairment. Unfortunately, the methodology of this study raised several concerns.

1. The inclusion criteria were poorly defined ie. cognitive dysfunction, middle-aged and older adults.

2. Why did studies reporting outcomes other than the cognitive incidence excluded?

3. The authors aimed to include case-control studies but specified that eligible studies must report the incidence of cognitive dysfunction. Case-control studies are not designed for studying incidence.

4. The purpose of conducting a systematic review is to identify and comprehensively synthesize all the relevant studies. Excluding studies with a Newcastle-Ottawa score of less than 4 might be considered inappropriate in the stage of the selection process. To avoid the effect of low-quality studies, the authors can perform sensitivity analysis instead of excluding the studies right from the beginning.

5. The authors did not specify the statistical model for the meta-analysis ie. DerSimonian and Laird or the weight model for the meta-analysis ie. inverse variance

6. Sensitivity analyses using the leave-one-out method are inappropriate. The authors should perform meaningful sensitivity analyses ie. excluding studies with a high risk of bias.

7. Egger's test was not designed to explore publication bias in meta-analyses with less than 10 studies.

8. The source of heterogeneity was insufficiently discussed.

9. Several grammatical errors were present.

Reviewer #2: The study question is clear and well defined, looking at the association between atopic dermatitis and cognitive dysfunction. Three databases were searched, Embase, Pubmed and Web of Science.

While the study aims to identify the potential mechanisms underlying the association between AD and cognitive dysfunction, the methods section does not describe any specific approaches for exploring these mechanisms, such as analyzing biomarkers or conducting neuroimaging studies. The authors may want to consider incorporating such approaches to gain a better understanding of the underlying potential mechanisms if they are going to describe it as one of the the aims.

More can be written on how this contributes to utility in clinical practice. More elaboration on the implications for the "prevention, diagnosis, and treatment of cognitive dysfunction in patients with AD".

Please kindly explain why the study Shang et al was used in this study since I could not find mentioning atopic dermatitis in the article.

The selected articles cover studies from United Kingdom, Taiwan, South Korea and Sweden. It should be mentioned that it may not be representative of all ethnicities and more focused on those in these countries. The "European" population mentioned in the study would come from these countries as well as the "Asian" analysed. It could be mentioned as a limitation and how it could further expand in future studies such as those in North America, other countries in Asian, Africa or Middle Eastern countries, if comparison in the results across ethnicities were to become more accurate and generalizable.

Overall, the article is almost ready to be accepted. Therefore, I recommend accepting this manuscript for publication with minor revisions to address the above issues.

Reviewer #3: This is an interesting study that focused on middle-aged and older adults and examined the association between atopic dermatitis and the risk of cognitive dysfunction.

The methodology and presentation of the authors' work are commendable, but establishing a causal relationship requires an empirical association, temporal priority of the independent variable, and non-spuriousness. Based on the available evidence and current data, it is challenging to definitively conclude that atopic dermatitis increases the risk of cognitive dysfunction.

additionally,

1) Are there any limitations regarding language when searching and selecting studies?

2) Authors should include detailed annotations that define each indicator used in the Newcastle-Ottawa Scale in Table S4.

3) Figure 3 is challenging to read, improvements should be made to enhance its readability.

4)P-values can never be 0. Please review the results in Figure 1, Supplementary Tables 4, 5, and 6 to ensure accurate reporting. If the exact p-value is less than 0.001, it is appropriate to state "p < 0.001". Otherwise, report the exact p-values.

5) The results from Figure 2 demonstrate high heterogeneity and imprecision. However, the potential causes of heterogeneity could not be fully explored.

6. PLOS authors have the option to publish the peer review history of their article (what does this mean?). If published, this will include your full peer review and any attached files.

Reviewer #1: No

Reviewer #2: No

Reviewer #3: No

---

## [Author Response · Author response to Decision Letter 0]

15 Aug 2023

Dear editor and reviewers,

Thank you for your letter dated July 12, 2023. On behalf of my colleagues, I am herewith submitting the revised manuscript (PONE-D-23-12015) entitled “Atopic dermatitis and cognitive dysfunction in middle-aged and older adults: A systematic review and meta-analysis” for consideration of publication in Plos one. We would like to thank the editor and reviewers’ work devoted to our manuscript and we are very grateful for their valuable suggestions. We have considered the comments carefully and have made revisions (highlighted in red in the revised manuscript with track changes ) which we hope meet with approval.

---

## [Decision Letter · Decision Letter 1]

12 Sep 2023

PONE-D-23-12015R1Atopic dermatitis and cognitive dysfunction in middle-aged and older adults: A systematic review and meta-analysisPLOS ONE

Dear Dr. Zhou, 

Thank you for submitting your manuscript to PLOS ONE. After careful consideration, we feel that it has merit but does not fully meet PLOS ONE’s publication criteria as it currently stands. Therefore, we invite you to submit a revised version of the manuscript that addresses the points raised during the review process.

ACADEMIC EDITOR: Your manuscript has been favorably reveiwed. Please find the reviewers' comments below and accordingly perform the mentioned minor revisions. 

We look forward to receiving your revised manuscript.

Kind regards,

Shuo-Yan Gau

Academic Editor

PLOS ONE

Journal Requirements:

Reviewers' comments:

Reviewer's Responses to Questions

**Comments to the Author**

1. If the authors have adequately addressed your comments raised in a previous round of review and you feel that this manuscript is now acceptable for publication, you may indicate that here to bypass the “Comments to the Author” section, enter your conflict of interest statement in the “Confidential to Editor” section, and submit your "Accept" recommendation.

Reviewer #2: All comments have been addressed

Reviewer #3: (No Response)

2. Is the manuscript technically sound, and do the data support the conclusions?

Reviewer #2: Yes

Reviewer #3: Yes

3. Has the statistical analysis been performed appropriately and rigorously? 

Reviewer #2: Yes

Reviewer #3: Yes

4. Have the authors made all data underlying the findings in their manuscript fully available?

Reviewer #2: Yes

Reviewer #3: Yes

5. Is the manuscript presented in an intelligible fashion and written in standard English?

Reviewer #2: Yes

Reviewer #3: Yes

6. Review Comments to the Author

Reviewer #2: The study has been well conducted with a few improvements that can be made that can make it a strong publication.

The inclusion criteria could be better defined, especially the definitions for cognitive dysfunction and middle-aged/older adults. For example, the age range or define the age cutoffs. For example for cognitive dysfunction, * you can refer to standard criteria or definitions used in published guidelines, such as the DSM-5 criteria for mild neurocognitive disorder or major neurocognitive disorder, or require that cognitive dysfunction was assessed through neuropsychological testing or clinical diagnosis, or specify the required threshold on cognitive screening tests like the Mini-Mental State Exam (MMSE) or Montreal Cognitive Assessment (MoCA).

Minor grammatical and language errors can be changed, such as "a thorough search of electronic databases". Or "Despite these findings, there was still a lack of consensus regarding the nature of the relationship between AD and cognitive dysfunction."

More attention can be given to the clinical and public health implications of the findings.

Reviewer #3: Most comments have been addressed well; however, I can't find your updated version of figures and tables.

7. PLOS authors have the option to publish the peer review history of their article (what does this mean?). If published, this will include your full peer review and any attached files.

Reviewer #2: No

Reviewer #3: No

---

## [Author Response · Author response to Decision Letter 1]

19 Sep 2023

1.Please review your reference list to ensure that it is complete and correct. If you have cited papers that have been retracted, please include the rationale for doing so in the manuscript text, or remove these references and replace them with relevant current references. Any changes to the reference list should be mentioned in the rebuttal letter that accompanies your revised manuscript. If you need to cite a retracted article, indicate the article’s retracted status in the References list and also include a citation and full reference for the retraction notice.

Response: We thank the reviewer for their valuable comments and suggestions.we have checked our reference list.

Comments to the Author:

Reviewer #2: The study has been well conducted with a few improvements that can be made that can make it a strong publication.

1.The inclusion criteria could be better defined, especially the definitions for cognitive dysfunction and middle-aged/older adults. For example, the age range or define the age cutoffs. For example for cognitive dysfunction, * you can refer to standard criteria or definitions used in published guidelines, such as the DSM-5 criteria for mild neurocognitive disorder or major neurocognitive disorder, or require that cognitive dysfunction was assessed through neuropsychological testing or clinical diagnosis, or specify the required threshold on cognitive screening tests like the Mini-Mental State Exam (MMSE) or Montreal Cognitive Assessment (MoCA).

Response: Thank you for your valuable feedback. We have revised our inclusion criteria as follows:We defined cognitive dysfunction as a category of cognitive impairment diseases that met the DSM-5 criteria for various subtypes of cognitive disorders, including all-cause dementia, Alzheimer's disease dementia, vascular dementia, mild cognitive impairment[1]. We defined middle-aged adults as those between 45 and 59 years old, while older adults as those aged 60 years and above[2]. 

References

[1]Majer R, Simon V, Csiba L, et al. Behavioural and psychological symptoms in neurocognitive disorders: Specific patterns in dementia subtypes[J]. Open Medicine, 2019, 14(1): 307-316.

[2] World Health Organization. (2020). Ageing and health. 

2.Minor grammatical and language errors can be changed, such as "a thorough search of electronic databases". Or "Despite these findings, there was still a lack of consensus regarding the nature of the relationship between AD and cognitive dysfunction."

Response: We thank for your constructive comments and suggestions. We have corrected the minor grammatical and language errors throughout the manuscript. For example, we have changed "a thorough search of electronic databases" to "a comprehensive search of electronic databases" in the Abstract section, and "Despite these findings, there was still a lack of consensus regarding the nature of the relationship between AD and cognitive dysfunction." to "However, these findings did not resolve the controversy over the molecular mechanism between AD and cognitive dysfunction." in the Introduction section.

3.More attention can be given to the clinical and public health implications of the findings.

Response: Thank you for your constructive comments and suggestions on our manuscript. We have revised our manuscript according to your requirements and added more discussion on the clinical and public health implications of our findings.We also acknowledged the limitations of our study and suggested directions for future research. 

Reviewer #3: Most comments have been addressed well; however, I can't find your updated version of figures and tables.

Response: We apologize for the inconvenience caused by the missing figures and tables in our previous submission. We have updated our manuscript with the revised and added figures and tables, which are now available in the supplementary materials. Thank you for your valuable feedback and suggestions.

---

## [Decision Letter · Decision Letter 2]

4 Oct 2023

Atopic dermatitis and cognitive dysfunction in middle-aged and older adults: A systematic review and meta-analysis

PONE-D-23-12015R2

Dear Dr. Zhou,

We’re pleased to inform you that your manuscript has been judged scientifically suitable for publication and will be formally accepted for publication once it meets all outstanding technical requirements.

Kind regards,

Shuo-Yan Gau

Academic Editor

PLOS ONE

Additional Editor Comments (optional):

Reviewers' comments:

Reviewer's Responses to Questions

**Comments to the Author**

1. If the authors have adequately addressed your comments raised in a previous round of review and you feel that this manuscript is now acceptable for publication, you may indicate that here to bypass the “Comments to the Author” section, enter your conflict of interest statement in the “Confidential to Editor” section, and submit your "Accept" recommendation.

Reviewer #2: All comments have been addressed

Reviewer #3: (No Response)

2. Is the manuscript technically sound, and do the data support the conclusions?

Reviewer #2: Yes

Reviewer #3: Yes

3. Has the statistical analysis been performed appropriately and rigorously? 

Reviewer #2: Yes

Reviewer #3: Yes

4. Have the authors made all data underlying the findings in their manuscript fully available?

Reviewer #2: Yes

Reviewer #3: Yes

5. Is the manuscript presented in an intelligible fashion and written in standard English?

Reviewer #2: Yes

Reviewer #3: Yes

6. Review Comments to the Author

Reviewer #2: The authors have made appropriate changes and improvements to the manuscript according to the comments and provided their response.

Reviewer #3: Please note that in your revised manuscript with track changes and some figures still report p-values as 0. Make sure to review the entire manuscript carefully and consider adjusting your analysis tool to generate more accurate figures before publication.

7. PLOS authors have the option to publish the peer review history of their article (what does this mean?). If published, this will include your full peer review and any attached files.

Reviewer #2: No

Reviewer #3: No

---

## [Editor Report · Acceptance letter]

12 Oct 2023

PONE-D-23-12015R2 

Atopic dermatitis and cognitive dysfunction in middle-aged and older adults: A systematic review and meta-analysis 

Dear Dr. Zhou:

I'm pleased to inform you that your manuscript has been deemed suitable for publication in PLOS ONE. Congratulations! Your manuscript is now with our production department. 

Kind regards, 

on behalf of

Mr. Shuo-Yan Gau 

Academic Editor

PLOS ONE